# ECHO: A Visio-Linguistic Dataset for Event Causality Inference via Human-Centric Reasoning

**Yuxi Xie    Guanzhen Li    Min-Yen Kan**
National University of Singapore
{xieyuxi, guanzhen}@u.nus.edu   knmnyn@nus.edu.sg

## Abstract

We introduce 🌀 ECHO (Event Causality Inference via Human-Centric Reasoning), a diagnostic dataset of event causality inference grounded in visio-linguistic social scenarios. ECHO employs real-world human-centric deductive information built on a television crime drama. ECHO requires the Theory-of-Mind (ToM) ability to understand and reason about social interactions based on multimodal information. Using ECHO, we propose a unified Chain-of-Thought (CoT) framework to assess the reasoning capability of current AI systems. Our ToM-enhanced CoT pipeline accommodates various large foundation models in both zero-shot and few-shot visio-linguistic reasoning. We use this framework to scrutinize recent large foundation models such as InstructGPT and MiniGPT-4 on three diagnostic human-centric tasks. Further analysis demonstrates ECHO as a challenging dataset to expose imperfections and inconsistencies in reasoning. Our data and code are publicly available at https://github.com/YuxiXie/ECHo.

## 1 Introduction

Social intelligence refers to the ability to understand, navigate, and respond effectively in social scenarios. As a widely investigated concept in psychology and social sciences, it has become a prominent facet of artificial intelligence (Walker and Foley, 1973; Kihlstrom and Cantor, 2000; Albrecht, 2006; Zadeh et al., 2019). In the development of social intelligence, humans gain the crucial cognitive capacity of understanding and reasoning about the mental states (*i.e.*, beliefs, desires, intentions, emotions, and thoughts) of individuals. This pivotal form of competence is commonly denoted as Theory-of-Mind (ToM) (Premack and Woodruff, 1978; Apperly and Butterfill, 2009; Apperly, 2010). As a fundamental ability of social commonsense reasoning (Davis, 2023), ToM is important for machine reasoning to achieve artificial general intelli-

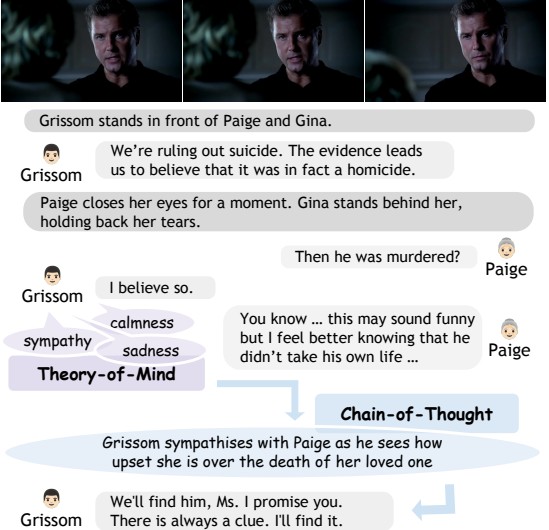

Figure 1: Scheme of our Theory-of-Mind enhanced Chain-of-Thought reasoning on human factors. We use scenes of key characters as visual representations.

gence (Goertzel, 2014; Zhong et al., 2023), especially towards better social intelligence.

Recently, large language (Chowdhery et al., 2022; Chung et al., 2022; Touvron et al., 2023; OpenAI, 2023) and multimodal (Radford et al., 2021; Alayrac et al., 2022; Li et al., 2022; Huang et al., 2023) models have exhibited remarkable reasoning capabilities. However, current large foundation models still fall short of adapting to personalized scenarios for specific users (Sap et al., 2022; Bubeck et al., 2023). Hence, there has been an increasing focus on human-centric reasoning as a means of enhancing artificial social intelligence for its integration in human daily life (Bard et al., 2020; Yuan et al., 2020; Moghaddam and Honey, 2023). To this end, ToM is one of the central challenges to accelerating communication and ensuring safety in human–computer interaction, requiring complex reasoning on how human beliefs and intents may vary across different scenarios (Yuan et al., 2020; Sap et al., 2022; Jin et al.,

2022; Sileo and Lernould, 2023). Specifically, current AI systems struggle with handling interleaved multi-modalities and faithful surmising for better consistency and interpretability of reasoning (Lyu et al., 2023; Bubeck et al., 2023).

To enhance reasoning consistency and faithfulness, recent works on large language models (LLMs) propose to break down a problem into intermediate inferences (Wei et al., 2022; Zhou et al., 2022; Xie et al., 2023). The impressive empirical success of this Chain-of-Thought (CoT) scheme on both textual and multimodal tasks (Wei et al., 2022; Zhang et al., 2023; Driess et al., 2023) demonstrates a promising paradigm to integrate ToM inference as an intermediate step in human-centric reasoning. In this work, we introduce ECHO (Event Causality Inference via Human-Centric Reasoning), a visio-linguistic dataset with ToM inferences for reasoning in social scenarios. With a focus on human factors, ECHO seeks to diagnose the social intelligence of current large language and multimodal models. We focus on event causality reasoning that remains challenging for recent LLMs (Kiciman et al., 2023).

We envision ECHO as a challenging diagnostic benchmark on human-centric reasoning. Each ECHO instance is grounded in a plot from the crime drama *CSI: Crime Scene Investigation*, enabling approximation of the real-world social interactions pertaining to the discordance in human beliefs. As shown in Figure 2, our core annotation process begins with ascertaining the identity of a specified character. Next, we discern their mental states via emotion interpretation. Leveraging this human-centric understanding, annotators then infer the cause or effect of a plot event for causality reasoning. To foster visio-linguistic social intelligence, we enhance ToM by guiding annotators to make causal inferences that take into account the mental states (*e.g.*, intentions, emotions, and thoughts) of characters and pinpoint related frames as visual evidence. As such, ECHO is integrated with a unified framework to assess human-centric reasoning in the social context. As detailed in Section 4, we propose a series of diagnostic tasks to evaluate capabilities to identify roles, reason about emotions, and infer event causality.

To conclude, we introduce ECHO, a challenging visio-linguistic corpus of human-centric reasoning in social scenarios. We propose a unified framework to evaluate existing large foundation models in zero-shot and few-shot ToM-enhanced CoT reasoning. Our further analysis demonstrates how to use our diagnostic tasks to assess multimodal understanding of human factors, revealing the deficiency of current AI systems in maintaining logical correctness and consistency throughout reasoning.

## 2 Related Work

ECHO takes a further step towards social intelligence on human-centric inference in visio-linguistic scenarios, probing the ToM capacity of large foundation models via CoT reasoning.

**Visio-Linguistic Reasoning.** Datasets and tasks in visio-linguistic reasoning span widely from descriptive information extraction (Antol et al., 2015; You et al., 2016; Gao et al., 2017), physical relation inference (Johnson et al., 2017; Hudson and Manning, 2019), to complex and deep reasoning on the event and human factors (Krishna et al., 2017; Zellers et al., 2019; Park et al., 2020). ECHO follows this trend to enhance the reasoning depth towards human-specific facets. Unlike recent works (Shen et al., 2020; **?**; Zhu et al., 2023c) of human-centric reasoning, ECHO is integrated with rigorous ToM annotations, supporting the final inferences via CoT reasoning for better consistency. Specifically, there are long-standing arguments on the development and assessment of ToM in both human psychology and machine intelligence (Premack and Woodruff, 1978; Apperly, 2010; Kosinski, 2023; Ullman, 2023). Measurement of ToM is usually based on *false belief* tasks (Dennett, 1978), assessing the ability to distinguish one's own (true) belief and others' (false) belief, given the information and experience asymmetry among different individuals. Constructed on crime drama, ECHO contains an abundance of such cases of *false belief* to probe ToM ability.

**Large Multimodal and Language Models.** Previous research in this area mainly complies with the paradigm of pre-training and fine-tuning to construct and train large-scale multimodal models to handle interleaved visual-and-linguistic information (Radford et al., 2021; Jia et al., 2021; Zellers et al., 2021; Alayrac et al., 2022; Li et al., 2022; Huang et al., 2023). Recently, there is the emergence of offline methods which leverage the capacities of large foundation models to conduct direct few-shot or zero-shot inference (Wu et al., 2023; Yang et al., 2023; Lu et al., 2023; Zhu et al.,

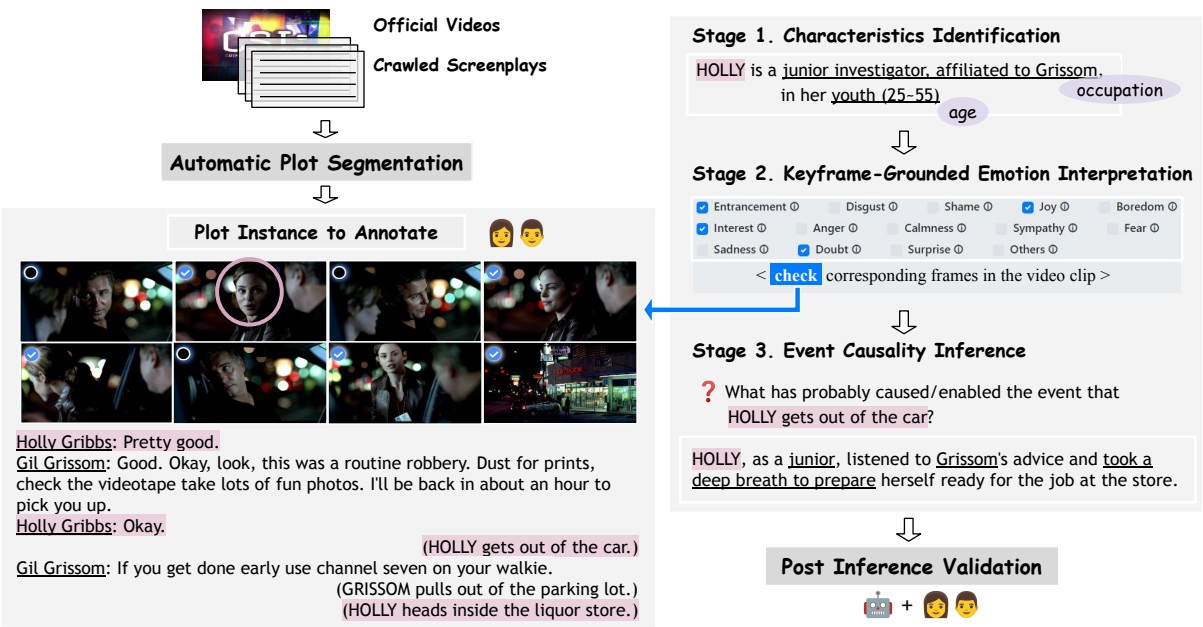

Figure 2: Annotation Pipeline of the dataset construction. We detail the second round annotation, where annotators provide ToM inferences in the first two stages following with the event causality inference.

2023a). Similar with Zhu et al. (2023a), we propose a framework to enhance visual understanding via LLM prompting (Ouyang et al., 2022; OpenAI, 2023) and facilitate LLM reasoning with augmented multimodal information (Li et al., 2023).

## 3 The ECHO Corpus

ECHO contains $2k$ inference instances collected via our ToM-enhanced CoT scheme. To facilitate the approximation of authentic social interactions, we ground ECHO in *CSI: Crime Scene Investigation*, an American procedural forensics crime series in English (Wikipedia contributors, 2023). With visual evidence and scenes in frames and utterances and narrations in screenplays, *CSI* represents a rich multimodal source, spanning widely factual, relational, and inferential data. As shown in Table 2, drama plots bring abundant instances of belief discrepancy and unexpected content for ToM reasoning. This enables our focus on human-centric information distillation and interpretation in ECHO's construction for ToM-enhanced inference.

### 3.1 Construction Pipeline

We pair official *CSI* clips with their screenplays, crawled from a publicly available website hosting TV show transcripts[1] (Frermann et al., 2018). We then launch annotation in 3 rounds with 30 annota-

tors working over 5 weeks after training sessions[2].

**Data Source Crawling and Preprocessing.** We acquire the official *CSI* videos with associated screenplays of 177 episodes from the first 8 seasons. We construct ECHO via further annotation and task formulation on a subset of *CSI* containing 15 episodes, each of which usually features one or two cases that are independent from the preceding plots. We develop heuristic rules[3] to automatically denoise, split, and categorize the scripts into plot events for subsequent task formulation.

**Round One: Plot Segmentation.** The crawled screenplays are distributed in discrete plots. We refine this segmentation with data cleaning to collect segments of feasible length and substantive contents. Different from previous works of automatic vision–language alignment (Myers and Rabiner, 1981; Frermann et al., 2018), we manually synchronize the screenplays with the time-stamped video clips to pinpoint the main characters for human-centric reasoning. We obtain $1,542$ plots grounded in different scenes in this round, with an average of 3 identified characters in each segment.

**Round Two: Inference Annotation.** Each annotation instance features one specified event for causality inference. We sequentially operationalize

---

[1] https://transcripts.foreverdreaming.org/

[2] Appendix A details annotator training.

[3] Details at https://github.com/YuxiXie/ECHo.

| # Clip | # Frame | | # Character | | # Inference | |
|---|---|---|---|---|---|---|
| | total | featured | total | labeled | cause | effect |
| 1,292 | 13,306 | 12,201 | 4,319 | 2,017 | 1,369 | 1,013 |

| Task | # Instance | # Annotation | Avg-Length |
|---|---|---|---|
| Role | 2,017 | 4,226 | 3.43 |
| Emotion | 2,017 | 4,271 | 2.44 |
| Event | 2,382 | 4,280 | 17.97 |

Table 1: Summative statistics of ECHO.

| # Clip | 100 | | |
|---|---|---|---|
| FB (44%) | 25 | 19 | 17 UC (36%) |
| C/E | cause (56%) | | effect (44%) |
| FB | objective (52%) | | subjective (48%) |
| UC | physical (69%) | | social (31%) |

Table 2: ToM attribute distribution on a 100-clip subset. **FB** and **UC** represent *false-belief* and *unexpected-content*, respectively.

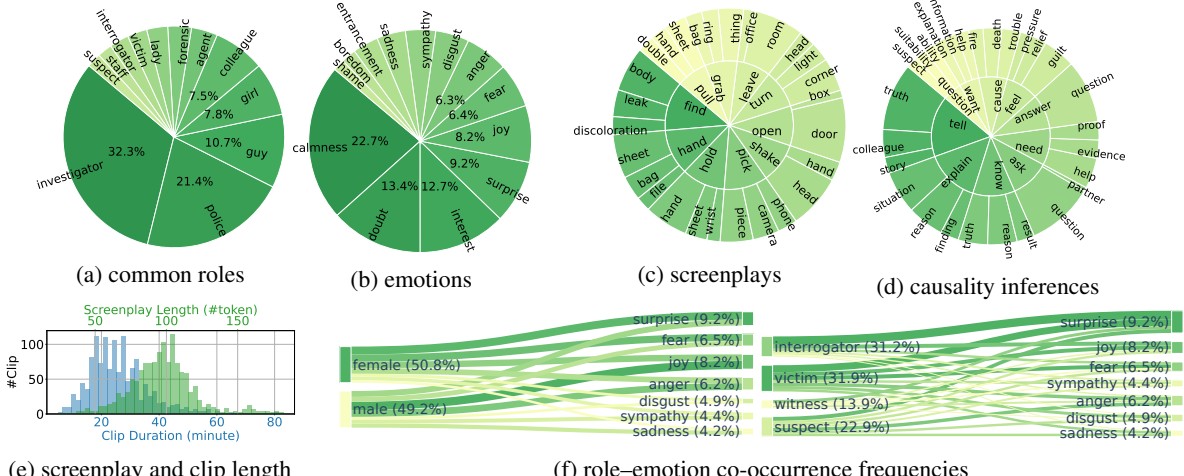

(a) common roles    (b) emotions    (c) screenplays    (d) causality inferences

(e) screenplay and clip length    (f) role–emotion co-occurrence frequencies

Figure 3: ECHO statistics and varieties of keywords. (a) and (b) show the distributions of the 12 most frequent roles and 13 emotions, respectively. (c) and (d) plot the top 10 most common root verbs (inner circle) and their top 3 direct noun objects (outer circle) in screenplays and event causality inferences, respectively. (f) demonstrates the coverage of role types associated with various emotions of equivalent proportions for a fair comparison.

annotation in 3 stages as follows:

1. *Characteristics Identification.* Given one key character, annotators identify the character's role in the plot. We encourage them to consider social attributes such as age range, occupation, and relations (with others) to describe the role. Character roles are not static and can vary in different plots, depending on the nuances of their appearances, behaviors, and interactions in the specific context.

2. *Keyframe-Grounded Emotion Interpretation.* We take a further step in human-centric reasoning to interpret mental states. Annotators choose from 13 primary emotions[4] categorized by adapting the 27 emotions from Cowen and Keltner (2017) to the crime drama. We also accept free-form input for emotions when no existing options apply. Annotators then extract associated frames that feature related emotions. We take the frames as visual representations of human factors. Specifically, when multiple emotions are identified, annotators are also instructed to select more frames. We do

not strictly enforce one-to-one matching between frames and emotions, since one frame can feature several emotions, and some emotions may be more accurately captured by considering the reactions of other characters. To ensure the completeness and informativeness of selected keyframes, we implement follow-up validation next in Round Three.

3. *Event Causality Inference.* Following the visio-linguistic human-centric inferences from the previous stages, we ask annotators to further infer the cause or effect of a specified event. We encourage them to consider the annotated roles and emotions to enhance reasoning consistency. Here we determine the events to annotate through two steps: 1) randomly select utterances or narrations that mention the main character(s) and occur in the middle of the plot to facilitate effective causality reasoning, and 2) filter out the automatically selected events that are insufficiently meaningful according to annotators' assessment.

We assign 2 to 4 annotators for each instance[5] for quality control. In this round, we collect a total of

---

[4]Including anger; boredom; calmness; disgust; doubt; entrancement; fear; interest; joy; sadness; shame; surprise; sympathy. We provide detailed definitions in Appendix C.

[5]For some instances, we additionally assign more annotators when the inter-agreement is low.

$5,746$ annotated instances. We use the ToM inferences to formulate diagnostic tasks in Section 4.

**Round Three: Inference Validation.** To qualify the annotated instances, we evaluate each data point via both automatic and manual checking. The real-time automatic checking alerts annotators if their inputs fail to meet certain informativeness criteria, such as text length and word-level overlap with the existing context. Our authoring team then carry out the manual validation. We particularly focus on instances with lower inter-rater agreement in emotion identification or anomalies in the timestamp distribution of selected frames. We specifically assess the plausibility, relevance, and completeness of annotations. For example, we reject instances where the event causality inference is weakly associated with either the plot or the annotated roles and emotions. Out of the $5,746$ annotations collected from Round Two, we finally retained $4,280$ annotations, as shown in Table 1.

### 3.2 Dataset Exploration

Table 1 and Figure 3e detail the summative statistics of collected data for the three tasks. The comparatively small scale of ECHO enables efficient assessment in the few-shot paradigm. With $4k$ ToM annotations on $2k$ inference instances, ECHO approximates authentic human-centric reasoning across visio-linguistic social scenarios. While our ECHO represents a subset of social interactions, we view it as an initial and specific step to explore ToM understanding of social intelligence. As outlined in Table 2, crime drama is rich in ToM-related cases, including *false-belief* and *unexpected-content*.

We further demonstrate the topic and scenario coverage of ECHO in Figure 3, by visualizing the keyword and verb–noun frequencies in both input screenplays and human annotations. Alone other lines, Figure 3f shows how the correlation between roles and emotions vary in ECHO, reflecting a closer alignment with the real-world social characteristics compared against other datasets. We analyze the potential bias in our data in Section 6.5.

## 4 Diagnose Human-Centric Reasoning in Visio-Linguistic Inference using ECHO

ECHO centers on rigorous human-centric information that supports ToM-enhanced CoT reasoning in visio-linguistic scenarios. We detail the formulation of our three sequential tasks to diagnose the human-centric reasoning ability next.

**Notation.** Each ECHO instance consists of a sequence of visual frames $V = [f_1, f_2, \cdots, f_N] = f_{1:N}$, a textual screenplay $T$. The frames are manually selected following the role and emotion identification in annotation. We gather frames featuring different characters together to represent the visual content of each clip. We designate each utterance or narration in the text to be an event $E_i$ for the causality inference, as $T = [E_1, E_2, \cdots, E_M]$. There is a key character $C$ to focus on in each instance.

**Task One: Role Identification** (*cf.* Annotation Round Two, Stage 1). The psychoanalysis of a person's role in social interactions indicates their identity, helping to infer their intentions, actions, and relations with others (Miller, 1962; Freese and Burke, 1994). Therefore, we test the ability of role identification to probe the fundamental human-centric understanding in ToM reasoning. Given frame(s) $f_i$ of the key character $C$ and the corresponding screenplay $T$, we prompt the model to generate the role $r$ of $C$. The role can be defined by age, occupation, or relations with others, as these attributes can exhibit a strong correlation with the human mental states for ToM reasoning.

**Task Two: Emotion Interpretation** (*cf.* Annotation Round Two, Stage 2). Emotions convey clues of mental states beyond verbal messages (Hari and Kujala, 2009). They bridge fundamental understanding (*e.g.*, role identification) and further inference (*e.g.*, intent prediction) in human-centric reasoning. We thus propose emotion interpretation as our second diagnostic task. We formulate this task as multi-choice question answering and test the alignment of model and human predictions on 13 candidate emotions, adapted from the taxonomy of Cowen and Keltner (2017) to the crime data.

**Task Three: Event Causality Inference** (*cf.* Annotation Round Two, Stage 3). Despite the practical success of large foundation models on a wide range of reasoning tasks, there is a debate as to whether they genuinely execute causal reasoning or just reproduce memorized patterns (Bender et al., 2021; Marcus, 2022). Furthermore, these models still produce imperfections such as erroneous logic and human-factor understanding (Ghazal et al., 2017; Bubeck et al., 2023; Zhong et al., 2023; Kiciman et al., 2023). Hence, we formulate a subsequent task as event causality inference to assess the causality reasoning capacity among socially-grounded events. We also utilize the ToM infer-

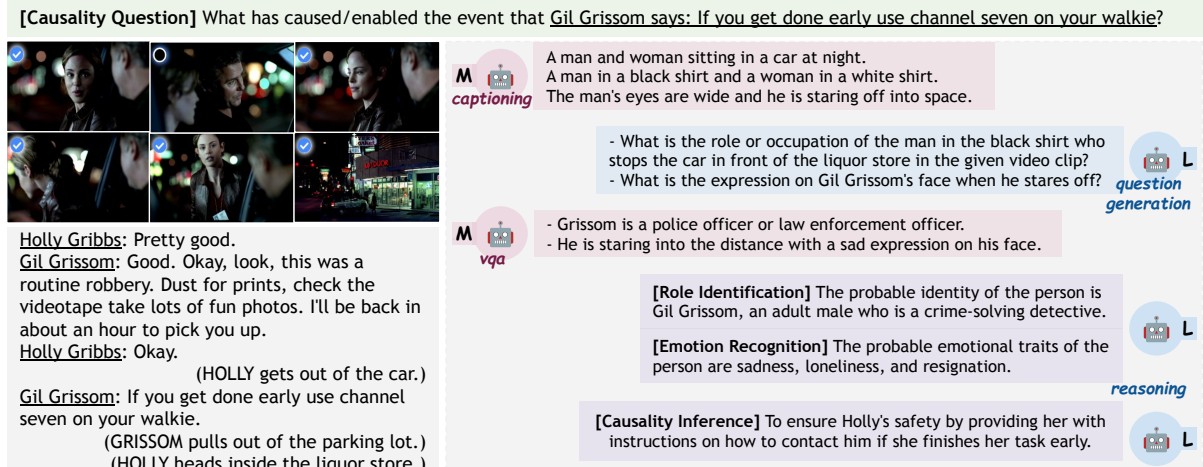

Figure 4: Examples of model inputs and outputs through the ToM-enhanced reasoning process. The language and multimodal models enhance understanding and reasoning of each other in a dialogue-like form.

ences from the preliminary tasks as the CoT intermediate step to test the reasoning consistency among intermediate ToM inferences and the final predictions. Specifically, with the frames $V$ and associated screenplay $T$, we ask models to infer the cause or effect of a given event $E_i$ in the context.

## 5 ToM-Enhanced CoT Reasoning

Given the inputs of visual frames $V$ and textual screenplay $T$, our objective is to make human-centric inference $I$. We follow Wei et al. (2022) to break down the process into intermediate steps $R$ and thus accommodate the three tasks in a unified framework to assess large foundation models.

As illustrated in Figure 4, our framework follows the Vision + LLM paradigm (Huang et al., 2023; Zhang et al., 2023; Yang et al., 2023) to facilitate multimodal understanding using the reasoning ability inherently grown in language models.

**LLM-Enhanced Multimodal Understanding.**
Enlightened by the advanced capability of LLMs in complex reasoning (Brown et al., 2020; Kojima et al., 2022; Chowdhery et al., 2022), there is an emergent line of research to leverage LLMs to prompt and guide information extraction in visual understanding (Surís et al., 2023; Wu et al., 2023; Yang et al., 2023; Zhu et al., 2023a). To diagnose the ability of human-centric reasoning of current large foundation models, we follow this paradigm to enhance multimodal information extraction with LLM reasoning. Specifically, we incorporate the LLM guidance as information-seeking questions to prompt multimodal understanding via visual

question answering. We simplify the framework of Zhu et al. (2023a) by directly generating one task-specific question instead of augmenting iterative questions with accumulated contextual information. Figure 4 demonstrates an example of using the LLM-generated question to enhance multimodal understanding for human-factor extraction.

**Vision-Augmented LLM Reasoning.** Reciprocally, the multimodal model can facilitate LLM reasoning by augmenting information grounded in the vision. To this end, the visual information should be projected into representations that LLMs can understand, such as discrete text words (Hu et al., 2022; Wang et al., 2022; Zeng et al., 2022; Yang et al., 2022) and continuous features adapted into the textual space (Tsimpoukelli et al., 2021; Alayrac et al., 2022; Driess et al., 2023; Huang et al., 2023; Li et al., 2023). In our framework, we follow the former line of work to supplement the multimodal model generated textual descriptions into the LLM for vision augmentation. Specifically, the visual information covers knowledge of various granularities, extracted by general captioning and task-specific question-prompted answering. The task-specific questions here are generated by the LLM to guide reasoning via ToM inference.

## 6 Experiments

We assess the social intelligence of existing large foundation models using the unified framework in Section 5 on our diagnostic tasks in Section 4. We ablate components in our framework on different backboned models and evaluate their effect on both

| Models | VL | FS | CoT | Role Identification | | | Emotion Interpretation | | |
|---|---|---|---|---|---|---|---|---|---|
| | | | | BleU-2 | Rouge-L | BERT-F1 | Macro-P | Macro-R | Macro-F1 |
| **BLIP-2** | ✗ | ✗ | ✗ | 1.80 | 7.86 | 45.46 | 23.42 | 27.67 | 23.52 |
| | ✓ | ✗ | ✗ | 2.35↑0.55 | 7.36↓0.50 | 43.96↓1.50 | 24.71 | 30.94 | 24.87↑1.35 |
| **MiniGPT-4** | ✗ | ✗ | ✗ | 0.95 | 7.13 | 44.39 | 24.75 | 6.66 | 6.56 |
| | ✓ | ✗ | ✗ | 0.47↓0.48 | 3.58↓3.55 | 41.21↓3.18 | 23.30 | 7.31 | 9.42↑2.86 |
| | ✗ | ✓ | ✗ | 2.52 | 10.43 | 49.33 | 26.99 | 8.44 | 6.05 |
| | ✓ | ✓ | ✗ | 0.69↓1.83 | 5.66↓4.77 | 40.13↓9.20 | 21.36 | 8.31 | 9.28↑3.23 |
| **InstructGPT** | ✗ | ✗ | ✗ | 0.95 | 5.08 | 44.10 | 23.38 | 43.64 | 29.76 |
| | ✗ | ✗ | ✓ | 1.57↑0.62 | 6.87↑1.79 | 46.21↑2.11 | 26.09 | 46.33 | 32.37↑2.61 |
| | ✓ | ✗ | ✗ | 2.57↑1.62 | 10.07↑4.99 | 49.21↑5.11 | 33.62 | 43.94 | 36.55↑6.79 |
| | ✓ | ✗ | ✓ | 2.07↑1.12 | 8.11↑3.03 | 46.89↑2.79 | 34.52 | 44.26 | 37.03↑7.27 |
| | ✗ | ✓ | ✗ | 3.16 | 10.48 | 49.79 | 24.29 | 37.61 | 28.93 |
| | ✗ | ✓ | ✓ | 3.04↓0.12 | 10.13↓0.35 | 50.07↑0.28 | 25.16 | 43.58 | 31.02↑2.09 |
| | ✓ | ✓ | ✗ | 6.48↑3.32 | 17.80↑7.32 | 53.43↑3.64 | 34.87 | 47.59 | 38.67↑9.74 |
| | ✓ | ✓ | ✓ | 5.79↑2.63 | 16.53↑6.05 | 53.37↑3.57 | 34.34 | 48.64 | 38.95↑10.02 |
| **Human (Inter-Annotator Agreement)** | | | | 9.34 | 18.93 | 57.12 | 85.00 | 67.71 | 75.36 |

Table 3: Result Comparison on Role Identification and Emotion Interpretation. **VL** indicates whether to input full screenplay or utilize frame-only information. **FS** and **CoT** represent few-shot and chain-of-thought, respectively.

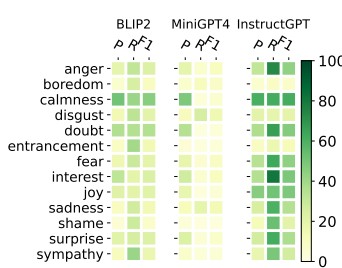

Figure 5: Class-wise performance comparison of three models on emotion interpretation.

| Models | VL | FS | CoT | ToM | BLeU-2 | Rouge-L | BERT-F1 |
|---|---|---|---|---|---|---|---|
| **MiniGPT-4** | ✗ | ✗ | ✗ | ✗ | 2.99 | 11.89 | 51.24 |
| | ✓ | ✗ | ✗ | ✗ | 3.30↑0.31 | 11.99↑0.10 | 51.62↑0.38 |
| | ✓ | ✗ | ✓ | ✗ | 3.31↑0.32 | 11.64↓0.25 | 51.21↓0.03 |
| | ✓ | ✗ | ✓ | ✓ | 3.55↑0.56 | 12.05↑0.16 | 51.98↑0.74 |
| **InstructGPT** | ✓ | ✗ | ✗ | ✗ | 7.44 | 18.41 | 59.33 |
| | ✓ | ✓ | ✗ | ✗ | 9.78↑2.34 | 21.30↑2.89 | 62.29↑2.96 |
| | ✓ | ✓ | ✓ | ✗ | 10.35↑2.91 | 22.02↑3.61 | 62.80↑3.47 |
| | ✓ | ✓ | ✓ | ✓ | 10.63↑3.19 | 22.20↑3.79 | 63.18↑3.85 |
| | ✓ | ✓ | ✓ | ✓ | 11.28↑3.84 | 22.97↑4.56 | 63.65↑4.32 |
| **Human (Inter-Annotator Agreement)** | | | | | 15.70 | 23.82 | 64.87 |

Table 4: Result Comparison on Event Causality Inference. **ToM** is the human-centric information. ✓(yellow) and ✓ represent model and human predictions, respectively.

automatic and human evaluation metrics.

## 6.1 Setup

**Backbones and Prompt Construction.** We use BLIP-2 (Li et al., 2023) and MiniGPT-4 (Zhu et al., 2023b), the recent public and reproducible multimodal models for visio-linguistic understanding. We evaluate InstructGPT (Ouyang et al., 2022) as the LLM backend considering the reproducibility of model performance, as stronger closed-source LLMs such as ChatGPT and GPT-4 (OpenAI, 2023) will be updated periodically. Details of prompt design can be found in Appendix E.

**Evaluation Metrics.** For generation tasks, we employ conventional metrics BLeU-2 (Papineni et al., 2002), Rouge-L (Lin, 2004), and BERTScore (deberta-xlarge-mnli) (Zhang et al., 2020). Considering the limitation of the automatic metrics capped at the reference quality (Zhu et al., 2023a), we conduct qualitative analysis to compare model and human predictions. For emotion interpretation as a multilabel classification, we adopt the macro precision, recall, and F1 scores as metrics. To fur-

ther validate whether the automatic metrics based on our annotated reference answers align with the actual quality of model predictions, we conduct additional human (on a subset – 238 instances – 10% of the whole set) and GPT-4 evaluation (on the whole set) on event causality inference results.

## 6.2 Results

We compare different models in zero-shot and few-shot settings. In event causality inference, we compare the impacts of CoT in different formats (indicated by "ToM"), including model-generated general intermediate steps (✗), model-generated ToM (✓), and human-annotated ToM (✓).

**Role Identification.** We see a huge gap between model and human generations in zero-shot prompting, while few-shot brings significant performance gain, especially on InstructGPT. This demonstrates the stronger ability of LLMs for in-context learning compared with MiniGPT-4. However, we observe a trend of performance drop when enhancing reasoning via CoT. This drop may be caused by the uncertainty due to task difficulty, leading to error

| Models | CoT | ToM | GPT-4 Score | Human Evaluation (Win / Lose) | | | |
|---|---|---|---|---|---|---|---|
| | | | | Plausibility | Relevance | Completeness | Overall |
| **InstructGPT** (Few-Shot) | ✗ | ✗ | 5.83 | − | − | − | − |
| | ✓ | ✗ | 5.90↑$_{0.07}$ | 7.1% / 5.7% | 43% / 31% | 38% / 26% | 46% / 28% |
| | ✓ | ✓ | 6.29↑$_{0.46}$ | 7.1% / 5.2% | 35% / 33% | **78**% / 12% | 56% / 25% |
| | ✓ | ✓ | 6.36↑$_{0.53}$ | **8.0**% / 4.7% | **50**% / 25% | 61% / 19% | **59**% / 20% |

Table 5: GPT-4 and human evaluation results on event causality inference.

accumulation in reasoning.

**Emotion Interpretation.** Likewise, InstructGPT shows a stronger in-context learning ability. Interestingly, MiniGPT-4 exhibits substantially poor performance on this task compared with the other models. We further diagnose this via the label-wise scores in Figure 5. The poor recall scores of MiniGPT-4 might be one of the reasons for the failure, as it tends to conduct single-label classification, neglecting the instruction in most cases. On the other hand, when sufficient information is provided, *i.e.*, with multimodal inputs and CoT reasoning, there is a stable increase in the F1 score. This disparity in how CoT works for role and emotion predictions may be attributed to the different degrees of uncertainty in reasoning for the two tasks. For example, models can resort to an expedient strategy to interpret emotions directly based on human facial expressions.

**Event Causality Inference.** We evaluate the effect of ToM-enhanced CoT reasoning on both multimodal and language models. For MiniGPT-4, basic CoT reasoning without a specified format or content of ToM cannot guarantee an improvement in performance. This is in accordance with our observation on role recognition that higher uncertainty may cause a performance drop. However, as demonstrated by the ToM-enhanced CoT reasoning, our proposed human-centric tasks can benefit the final inference by incorporating ToM information about roles and emotions. Furthermore, despite the incompleteness of labeled human predictions (as shown in Table 1), human ToM still exhibits a more significant effect in the reasoning process.

### 6.3 Ablation Study

We conduct further analysis to probe the impact of different modalities and ToM-CoT reasoning.

**Vision vs. Language Models.** In the diagnostic tasks, we observe large differences in the generations between multimodal and language models. As the performance drops remarkably when in-

corporating textual information into multimodal models such as MiniGPT-4, we see that these multimodal models still struggle to handle long input contexts. This demonstrates the importance of LLM incorporation for multimodal understanding to enhance and guide information extraction and deduction for further reasoning.

**ToM-enhanced CoT Reasoning.** The performance gain from CoT reasoning in Table 4 varies when incorporating intermediate inferences from different sources, where human-annotated ToM (partially labeled) still outperforms the others. This demonstrates the deficiency of current large foundation models in eliciting and utilizing the ToM inferences for better reasoning.

### 6.4 Qualitative Evaluation

We compare the InstructGPT-based model with the CoT and/or ToM mechanisms in reasoning against the vanilla version that only uses few-shot prompting in human evaluation. We choose the criteria including plausibility, relevance, completeness, and overall quality and let the evaluators judge which one is better. Table 5 shows the human evaluated win/lose rate on 10% (238 instances) of the whole dataset on event causality inference. To obtain a more complete understanding, we conduct GPT-4 evaluation on the whole set. Specifically, we adapt the scoring framework from Zheng et al. (2023), with our annotated inferences as the reference answers, where scores range from 1 to 10.

We observe the same trend of model performance on human judgment and GPT-4 scoring in terms of the overall quality of predictions. For example, the CoT framework with human-annotated ToM achieves the highest GPT-4 score and win rate at 6.36 and 59%, respectively. However, we see different trends in the relevance and completeness ratings, where the model-generated ToM-enhanced CoT achieves the best completeness but worst relevance scores. This may come from the LLM abilities of information extraction and understanding which, however, also brings hallucination.

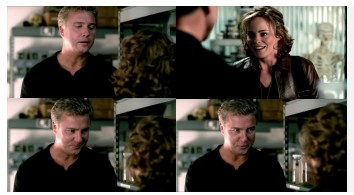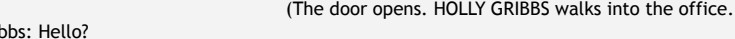

Holly Gribbs: Hello?

(The door opens. HOLLY GRIBBS walks into the office.)

(She looks around and grimaces at the various items on the shelves.)
(Behind her, GRISSOM walks up to her.)

Gil Grissom: Hi.

(HOLLY is startled. She gasps and turns around.)

Gil Grissom: Sorry. Welcome to Forensics. Gil Grissom. I'm your supervisor on graveyard.
Holly Gribbs: Holly Gribbs.

[Event Causality Inference] What probably has caused/enabled the event that Behind her, GRISSOM walks up to her?
[Human] Grissom is working at the office at Forensics, likely on his shift, when Holly, a new employee, enters.
He knows she is going to report today, and is ready to greet her.

[Role & Emotions] Grissom is supervisor in middle age, who may have the feelings of surprise, calmness, slightly sorry, and appreciation.
Holly is new employee at Graveyard in youth, who may the feelings of disgust, doubt, fear, and surprise.
[(Hum-ToM) MiniGPT-4] Grissom has been waiting for Holly to walk into the office.
[(Hum-ToM) InstructGPT] Holly has entered the room and Grissom noticed her presence. Grissom has curiosity and interest in meeting her.
[CoT (MiniGPT-4)] Grissom is a man in black leather jacket, he was wearing a dark blue shirt.
Holly's body language changes when Grissom walks up to her.
[MiniGPT-4] Holly Gribbs is in front of Grissom wearing black from behind.
[Role & Emotions (InstructGPT)] Holly Gribbs is a new employee in the office, who may have the feelings of surprise, curiosity, and
nervousness in the clip. Grissom is her supervisor, who may have the feelings of professionalism, and confusion in the clip.
[InstructGPT] Grissom is introducing himself to Holly Gribbs, as a form of greeting and to establish a professional relationship.

Figure 6: We compare the ToM-enhanced inferences between human annotators and different models. We consider three sources of the incorporated ToM information, including human, MiniGPT-4, and InstructGPT generations, as shown in green, pink, and blue, respectively. Imperfections are highlighted in yellow.

## 6.5 Discussion

We discuss our main findings in qualitative analysis by answering the following questions:

**Q1. Can models maintain reasoning robustness when input information varies in format?**
As shown in Figure 6, LLMs present significantly higher adaptiveness to elicit different input information for reasoning. For example, InstructGPT directly synthesizes the character emotional traits such as "curiosity" in the human-annotated ToM-enhanced inference, while MiniGPT-4 is still at copy-and-paste level in text generation and tends to focus more on descriptive information in vision.

**Q2. Can models maintain consistency and faithfulness throughout ToM-CoT reasoning?**
MiniGPT-4 shows fact-level consistency in inference via reiterating or rephrasing selected spans. However, it struggles in reasoning about implicit or intermediate information. On the other hand, despite the advancement of LLMs, they may produce problematic hallucination, *i.e.*, imperfect predicted ToM such as "confusion" can lead to wrong final inference that may totally contradict the fact.

**Q3. What potential bias exists in ECHO that can lead to erroneous model predictions?**
One crucial problem we find in the MiniGPT-4 outputs is that it tends to randomly check one emotion option when it is not confident in the selection. This indicates a high uncertainty in the multimodal model in the mental state interpretation of humans. Possible reasons can come from both the model

and data sides. Specifically, information from still frames can cause ambiguity without clip details, as shown by the erroneous "confusion" in Figure 6. On the other hand, we acknowledge that our dataset ECHO represents a subset of social interactions, but we view it as an initial and specific step to explore ToM understanding of social intelligence.

## 7 Conclusion

We introduce a visio-linguistic dataset ECHO to probe human-centric social intelligence. With our ToM-enhanced CoT framework, we diagnose the reasoning ability of large foundation models. Experiment results and further analysis demonstrate the deficiency of current AI systems and potential bias in ECHO for efficient, correct, and consistent reasoning. We foresee follow-up work on both model and data facets to develop faithful reasoning across a broader range of social scenarios.

## Limitations

**Dataset Scale and Generalizability.** As shown in Table 1 and Section 6.5, there can be potential bias in ECHO since we only label half of the featured characters to reason about their ToM. This imbalance of human belief considerations can lead to bias in final inferences as models may only focus on the thoughts and intentions of some of the characters. Furthermore, the reliance on crime scene content may restrict its applicability to a specific genre related to crime instead of daily life scenarios. While our ToM annotations (*e.g.*, emotions)

can represent human's mental states in daily life, future work may further explore whether and how ToM inferences in different distributions can assist human-centric reasoning in more general scenarios.

**Dataset Construction.** We may lack a detailed analysis of the visual representations to demonstrate how this information complements the textual inputs. In event causality inference, we adopted automatic event determination to ease the efforts of human annotation, which cannot prioritize salience within the plot. This means that certain events, potentially more interesting or pertinent for causal reasoning, may go unselected.

In future work, we will further refine ECHO to validate and extend visio-linguistic ToM inferences to improve the coverage and balance of event topics, reasoning types, and source of inference evidence.

## Ethics Statement

We have received approval from the Institutional Review Board (IRB)[6] for our data annotation. We design the training tutorial and experimental sessions as guided and reviewed by the IRB to maintain minimal risks to participants. The review process took two months to complete.

Since ECHO contains criminal data with violent content, it may enable malevolent imitation actors or harm to specific groups of people. To avoid this misuse potential of ECHO, we will impose strict rules for access requirements and frequently track the follow-up works to constrain its usage within research-only goals. In the future, we will also make regular updates on ECHO to further extend and balance the ToM attributes to alleviate potential bias and ambiguity in datapoints for better generalizability of our diagnostic tasks.

## Acknowledgements

The computational work for this paper was partially performed on resources of the National Supercomputing Centre (NSCC), Singapore[7]. We would like to thank our annotators for their time and efforts in annotating and validating ECHO instances and for our pilot subjects for their insightful feedback for our annotation user interface refinement.

---

[6]https://www.nus.edu.sg/research/irb.
[7]https://www.nscc.sg/

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

# A    Annotator Recruitment and Training

We recruited prospective annotators considering both quality and diversity. First, all selected annotators possess a minimum of undergraduate education and demonstrate familiarity or expertise with the TV series *CSI* to ensure the annotation quality. Second, to mitigate potential biases, our recruitment aimed for a balanced distribution across various factors, including gender, academic disciplines, and nationalities.

We piloted our preliminary annotation pipeline for refinement. After subsequent refinement, prospective annotators are first asked to watch a prepared instruction video[8] and complete a pre-annotation quiz to demonstrate their understanding of tasks. Our team manually checked applicants' responses for quality, admitting qualified subjects as participants. We show one example quiz question as follows.

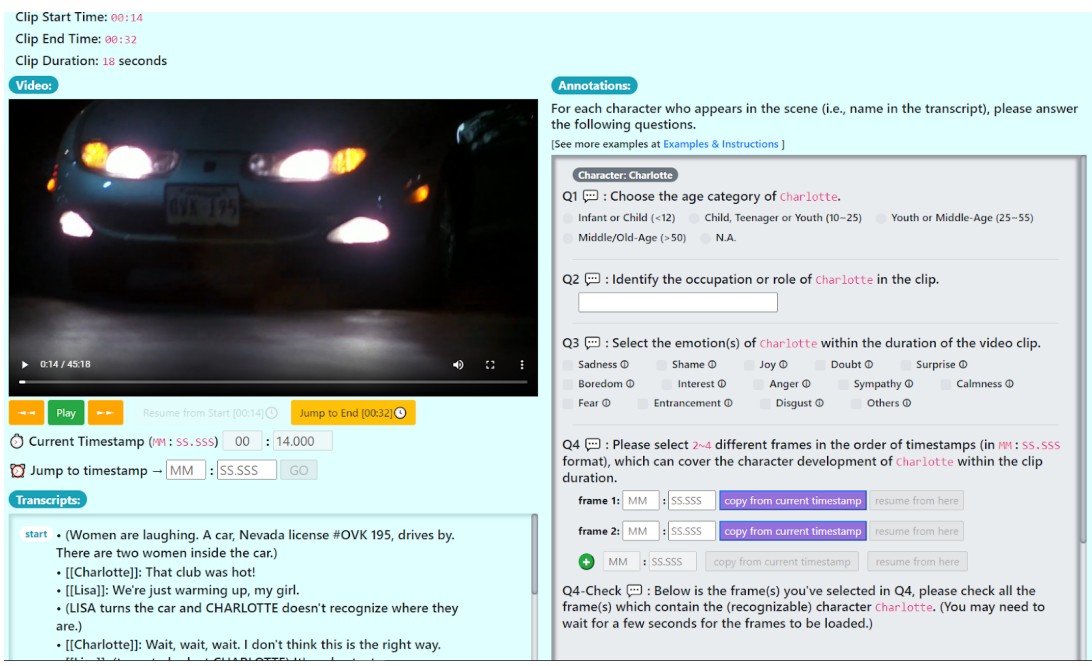

**General Qn.** Select the important elements to focus on for each sample.
☐ a short video clip with the provided start and end times
☐ the whole video clip of the episode
☐ transcripts of the clip
☐ align characters in the clip with their names in the transcripts
☐ the key character to focus on for all the annotation questions
☐ different characters to focus on for different annotation questions

**Q2-Related Qn.** Select the options which can be the input to Q2.
☐ occupation (for a living), e.g. singer, police, officer
☐ role (role in the event), e.g., driver, customer, suspect of the crime
☐ role (relation with others), e.g., the woman who stares at the others
☐ appearance description, e.g., the girl in a white shirt

**Q3-Related Qn.** What kind of emotions should be selected, and what is "Others" for?
☐ some emotions appearing via the key character's facial expressions/actions
☐ all emotions appearing via the key character's facial expressions/actions
☐ possible emotions of the key characters reflected by the others they interact with
☐ "Others": to add emotions that aren't included in the options

**Q4-Related Qn1.** What kind of & How many frames should be selected?
☐ all the frames indicating the change of (emotional, motional) states of the key characters
☐ the frames should preferrably be evenly distributed on the clip
☐ there is no lower bound of the the selected number

---

[8] https://vlcsr.comp.nus.edu.sg/static/video/VL_event_causality_annotation_instruction.mp4

☐ there is no upper bound of the the selected number

**Q4-Related Qn2.** Requirements for the frames to be checked.

☐ contain the facial expressions / recognizable actions of the key character

☐ contain only external information (others reflection) which may indicate the state of the key character

**Q5-Related Qn.** Select the requirements for the input text.

☐ Keep a low overlap-rate with the transcripts (< 30%)

☐ Start with the text provided in front of the text box

☐ Focus on the key character

☐ Can consider emotion change of the character as labeled in Q3

☐ Try to elaborate on the intrinsic logic among events which aren't directly described in the clip/transcripts

## B  Annotation Interface

We provide a publicly available webpage[9] to demonstrate the example annotations on an instance in Round Two. Below is an example of the detailed annotation questions.

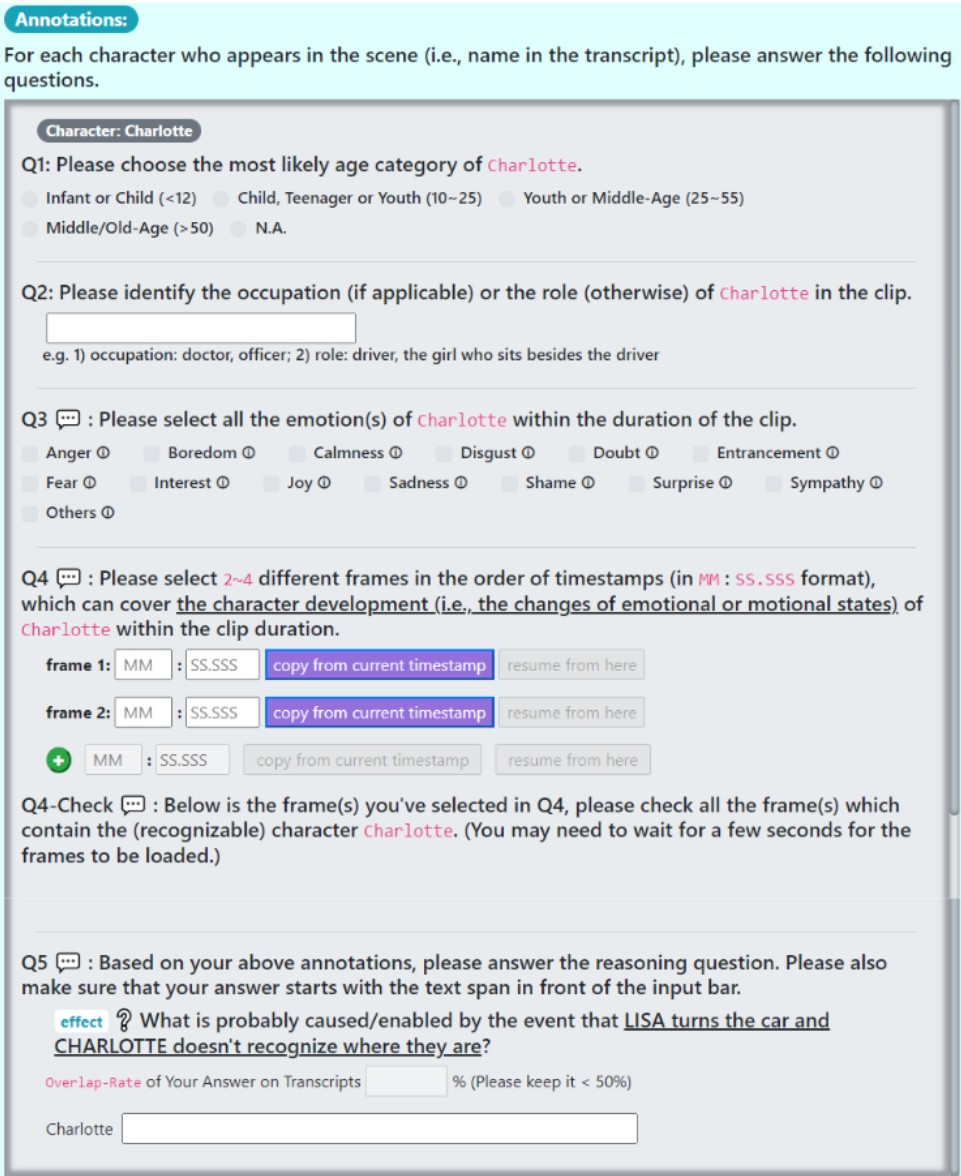

---

[9] https://yuxixie.github.io/_pages/CSI_example.html

## C  Emotion Categorization

We adapt the original 27 emotions from Cowen and Keltner (2017) to crime plots, merging them into 13 emotions more suited for the domain:

- **Anger**: wrath, outrage, fury, violence, irritability, hostility, resentment;
- **Boredom**: blahs, doldrums, ennui, weariness, listlessness, restlessness;
- **Calmness**: blahs, doldrums, ennui, weariness, listlessness, restlessness;
- **Disgust**: contempt, scorn, disdain, aversion, distaste, revulsion;
- **Doubt**: uncertainty, confusion, distrust;
- **Entrancement**: brooding, reverie, contemplation, daydreaming, cogitation, detachment;
- **Fear**: anxiety, dread, fright, panic, nervousness;
- **Interest**: trust, kindness, affection, devotion, acceptance, love, anticipation, friendliness;
- **Joy**: enjoyment, bliss, happiness, relief, delight, pride, thrill, ecstasy;
- **Sadness**: grief, sorrow, gloom, despair, melancholy, loneliness, depression;
- **Shame**: regret, guilt, embarrassment, remorse;
- **Surprise**: astound, shock, astonishment, wonder;
- **Sympathy**: commiseration, compassion, feeling.

## D  Frequently Asked Questions

### D.1  The fact that multiple different causes of an event can exist complicates the evaluation of the CoT approach for event causality. How is evaluation impacted when multiple causes of an event may exist?

We dealt with the one-to-many problem on two aspects of data collection. First, we provide multiple reference inferences (annotations) for each event, as shown in Table 1. Second, we narrow the scope of potential causes by employing Theory-of-Mind (ToM) as the intermediate reasoning step. This serves as a directional constraint to reduce the search space of potential CoT chains.

Considering that our annotated reference answers may not capture the full spectrum of possible causes, we conduct both human and GPT-4 evaluations to provide qualitative analysis on the model predictions. The human evaluation results are consistent with the automatic metrics assessed in Table 4. Furthermore, the performance gain brought by ToM-constrained CoT shows the importance of ToM in our human-centric reasoning task.

### D.2  How many instances are rejected and revised during inference validation? What are the major error types and feedback types encountered during annotation and verification?

Out of the $5,746$ annotations collected from Round Two, we finally retained $4,280$ annotations, as indicated in Table 3. Among these, $107$ were revised and accepted following re-annotation. In validation, we check the annotation quality considering three criteria: plausibility, relevance, and completeness. Specifically, we mainly reject instances if they exhibit weak connections between the annotated inferences and the plot contents or the corresponding character roles and emotions.

Feedback from annotators suggests two primary causes for annotation errors: 1) the imperfect event selection (where we reject the entire instance), and 2) insufficient incorporation of ToM in reasoning (where we reject outright or send it back for re-annotation).

### D.3  Why use instructGPT instead of stronger LLMs like ChatGPT and GPT-4?

We use InstructGPT considering the reproducibility of model performance, as stronger closed-source LLMs like ChatGPT and GPT-4 will be updated periodically. For reference, we conduct an additional experiment using ChatGPT (`gpt-3.5-turbo-0613`) in Table 6 on the task of event causality inference.

## E  Implementation Details

We detail our experiment setup and prompt construction in this section.

| Models | VL | FS | CoT | ToM | BERT-F1 | GPT-4 Score |
|---|---|---|---|---|---|---|
| InstructGPT | ✓ | ✓ | ✓ | ✓ | 63.18 | 6.29 |
| | ✓ | ✓ | ✓ | ✓ | 63.65 | 6.36 |
| ChatGPT | ✓ | ✓ | ✓ | ✓ | 64.09 | 6.98 |

Table 6: Result comparison on InstructGPT and ChatGPT.

## E.1 Setup

For multimodal models, we use BLIP-2[10] (blip2_t5) (Li et al., 2023) and MiniGPT-4[11] (prerained_minigpt4_7b) (Zhu et al., 2023b), the recent public and reproducible multimodal models for visio-linguistic understanding. Specifically, we use pretrain_flant5xl for BLIP-2 due to the computation limit. For LLM backend, we use InstructGPT (text-davinci-003) (Ouyang et al., 2022).

In the automatic evaluation of text generation tasks, we measure the similarity between model outputs and human annotations, compared with the inter-agreement among annotators using the same metrics. For the multilabel classification task, the macro precision, recall, and F1 scores measure both the model–human and inter-annotator agreement. The inter-annotator agreement considers the instance-wide similarity between one annotator and the others.

## E.2 Pipeline Construction

Denote the multimodal model and the language model as $\mathcal{M}$ and $\mathcal{L}$ respectively. The process for ToM-enhanced CoT reasoning for event causality inference is as follows (Steps 2 and 3 can be merged for simplicity, with further post-processing to extract the final answers):

**Step 1: Visual Information Extraction using $\mathcal{M}$.** $\mathcal{M}$ generates visual descriptions based on the input video clips. The prompt to $\mathcal{M}$ determines the type and focus of the visual descriptions generated.
- input: clip frames
- output: visual descriptions

**Step 2: Role and Emotion Identification using $\mathcal{L}$.** We augment the text prompt for $\mathcal{L}$ with the visual information obtained previously, eliciting roles and emotions associated with given characters.
- input: visual descriptions + textual context (screenplay)
- output: roles and emotions of the specified characters

**Step 3: ToM-Enhanced CoT Reasoning using $\mathcal{L}$.** Using the predicted roles and emotions as intermediate reasoning products, we construct the CoT prompt for $\mathcal{L}$, allowing it to perform ToM-enhanced reasoning about event causality.
- input: visual descriptions + textual context (screenplay) + roles and emotions
- output: event causality inference

## E.3 Prompt Design

For multimodal prompting, we adopt the form of visual question answering where specific instructions will be provided for human-centric information extraction and inference. For LLM prompting, we accommodate information comprising screenplays, textual descriptions of visual frames in multi-granularities, and specific instructions to stimulate reasoning.

---

[10] LAVIS: https://github.com/salesforce/LAVIS
[11] MiniGPT-4: https://minigpt-4.github.io/

| **ROLE** |
| --- |
| The image is a video frame of a person. |
| Specifically, the role of a person can be their occupation or their relation with others. |
| In this case, what probably is the role of this person? |
| Answer: |
| **EMOTION** |
| The image is a corresponding video frame of a person. |
| Question: What are the possible emotional traits of this person? |
| Answer Choices (can select more than one choices): |
| {#1} |
| Answer: |

Table 7: Prompts for BLIP-2 with vision-only inputs. #1 represents the emotion options to choose from.

| **ROLE** |
| --- |
| Read the screenplay of a video clip as follows: |
| {#1} |
| For reference, the image is a corresponding video frame of a person. |
| Specifically, the role of a person can be their occupation or their relation with others. |
| In this case, what probably is the role of this person? |
| Answer: |
| **EMOTION** |
| Read the screenplay of a video clip as follows: |
| {#1} |
| For reference, the image is a corresponding video frame of a person. |
| Question: What are the possible emotional traits of this person? |
| Answer Choices (can select more than one choices): |
| {#2} |
| Answer: |

Table 8: Prompts for BLIP-2 with vision-and-language inputs. #1 represents the screenplay content.

| **ROLE** |
| --- |
| Below are frames of a person happening in chronological order: |
| {#1} |
| Specifically, the role of a person can be their occupation or their relation with others in the plots. |
| In this case, what probably is the role of this person? |
| Answer: |
| **EMOTION** |
| Below are frames of a person happening in chronological order: |
| {#1} |
| Question: What are the possible emotional traits of this person? |
| Answer Choices (can select more than one choices): |
| {#2} |
| Answer: |
| **EVENT CAUSALITY** |
| Below are several frames (with characters they contain) happening in chronological order in the video clip: |
| {#1} |
| What is the possible {#3} of the clip event that {#4}? |
| Answer: |

Table 9: Prompts for MiniGPT-4 with vision-only inputs. #1 represents the visual tokens as a placeholder for frame embeddings. #3 and #4 represent the causality type (cause or effect) and the event to reason about, respectively.

| | |
|---|---|
| **ROLE** | |

Read the screenplay of a video clip as follows:
{#1}
For reference, below are a series of chronologically ordered frames of the person {#2} from the same video clip:
{#3}
What probably is the role of this person?
Answer:

**EMOTION**

Read the screenplay of a video clip as follows:
{#1}
For reference, below are a series of chronologically ordered frames of the person {#2} from the same video clip:
{#3}
Question: What are the possible emotional traits of this person?
Answer Choices (can select more than one choices):
{#4}
Answer:

**EVENT CAUSALITY**

Read the screenplay of a video clip as follows:
{#1}
For reference, below are a series of chronologically ordered frames from the same video clip, with certain frames annotated to indicate the character(s) they feature:
{#3}
What is the possible {#4} of the clip event that {#5}?
Answer:

Table 10: Prompts for MiniGPT-4 with vision-and-language inputs. #2 represents the name of the specified character.

**VQA**

Below are a series of chronologically ordered frames from a video clip, with certain frames annotated to indicate the character(s) they feature:
{#1}
Based on the provided frames, please answer the following question.
Question: {#2}

Table 11: Prompts for MiniGPT-4 for clip-based visual question answering.

**ROLE**

Below is the description of a video clip where there is a character named {#1}:
{#2}
Specifically, the role of a character can be their occupation or their relation with others in the plots.
In this case, what is the role of {#1}?
Answer:

**EMOTION**

Below is the description of a video clip where there is a character named {#1}:
{#2}
Question: What are the possible emotional traits of {#1}?
Answer Choices (can select more than one choices):
{#3}
Answer:

**EVENT CAUSALITY**

Below is the description of a video clip where there is a character named {#1}:
{#2}
What is the possible {#3} of the clip event that {#4}?
Answer:

Table 12: Prompts for InstructGPT with vision-only inputs.

**ROLE**

Read the screenplay of a video clip as follows:
{#1}
Specifically, the role of a character can be their occupation or their relation with others in the plots.
In this case, what is the role of {#2}?
Answer:

**EMOTION**

Read the screenplay of a video clip as follows:
{#1}
Question: What are the possible emotional traits of {#2}?
Answer Choices (can select more than one choices):
{#3}
Answer:

**EVENT CAUSALITY**

Read the screenplay of a video clip as follows:
{#1}
What is the possible {#2} of the clip event that {#3}?
Answer:

Table 13: Prompts for InstructGPT with language-only inputs.

**ROLE**

Read the screenplay of a video clip as follows:
{#1}
For reference, below is the description of the same video clip:
{#2}
In this context, a character's role can be defined by their occupation or their relationships with others within the plots.
In this case, what is the role of {#3}?
Answer:

**EMOTION**

Read the screenplay of a video clip as follows:
{#1}
For reference, below is the description of the same video clip:
{#2}
Question: What are the possible emotional traits of {#3}?
Answer Choices (can select more than one choices):
{#4}
Answer:

**EVENT CAUSALITY**

Read the screenplay of a video clip as follows:
{#1}
For reference, below is the description of the same video clip:
{#2}
What is the possible {#3} of the clip event that {#4}?
Answer:

Table 14: Prompts for InstructGPT with vision-and-language inputs.

| |
| --- |
| **ROLE** |
| Read the screenplay of a video clip as follows: |
| {#1} |
| For reference, below is the description of the same video clip: |
| {#2} |
| In this context, a character's role can be defined by their occupation or their relationships with others within the plots. |
| As an intermediate step to discern the role of {#3}, we may need to delve deeper into specific and concrete visual cues from the video clip. |
| Assuming you can access the video clip, please generate a question that indirectly seeks information about the role of {#3}. |
| Question: |
| **EMOTION** |
| Read the screenplay of a video clip as follows: |
| {#1} |
| For reference, below is the description of the same video clip: |
| {#2} |
| As an intermediate step to interpret the emotional trait(s) of {#3}, we may need to delve deeper into specific and concrete visual cues from the video clip. |
| Assuming you can access the video clip, please generate a question that indirectly seeks information about the emotion(s) of {#3}. |
| Question: |
| **EVENT CAUSALITY** |
| Read the screenplay of a video clip as follows: |
| {#1} |
| For reference, below is the description of the same video clip: |
| {#2} |
| In order to understand the possible {#3} of the clip event that {#4}, we may need to delve deeper into specific and concrete visual cues from the video clip. |
| Assuming you can access the video clip, please construct a question that indirectly seeks information useful for inferring the causality of the event. |
| Question: |

Table 15: Prompts for InstructGPT with vision-and-language inputs for information-seeking question generation.