# OpenReview forum: "ECHo: A Visio-Linguistic Dataset for Event Causality Inference via Human-Centric Reasoning"
_EMNLP/2023/Conference — EMNLP 2023 Findings_

### Official Review · Reviewer_CzKR · 2023-08-04

**Soundness:** 4

**Excitement:**

3: Ambivalent: It has merits (e.g., it reports state-of-the-art results, the idea is nice), but there are key weaknesses (e.g., it describes incremental work), and it can significantly benefit from another round of revision. However, I won't object to accepting it if my co-reviewers champion it.

**Paper Topic And Main Contributions:**

This paper presents ECHo, a diagnostic dataset focused on event causality inference grounded in visio-linguistic social scenarios. The dataset is carefully collected from CSI, a crime drama, incorporating video and screenplay information. It includes three annotation tasks: Characteristics Identification, Keyframe-Grounded Emotion Interpretation, and Event Causality Inference. The study also evaluates the performance of LLM and multimodal models in these tasks, demonstrating that a ToM-enhanced model shows improved reasoning capabilities.

----
Thanks for the detailed response.

**Questions For The Authors:**

A. In Characteristics Identification, are different roles of the same character collected for each plot, or does a character's role remains consistent across most plots?

B. In Keyframe-Grounded Emotion Interpretation, if multiple selections are allowed, how do annotators choose key frames for each emotion selection?

C. In Event Causality Inference, are there any limitations on using utterances or narrations as events? How the selection of annotated "events" is determined?

D. Sec 5 and Fig 4 are hard for me to understand. It can be helpful to clarify what intermediate steps are involved, what is the input and output of each step, and which model and prompt are used. There are too many prompts in Appendix D and it's hard for me to map them to each intermediate step. I also don't understand why some prompt requires a character name but some do not. How does the model know which person should be focused on?

E. How to get the visual frames V in Sec 5? Are they provided by the annotated dataset or automatically selected from a video?

F. How many instances are rejected and revised during inference validation? What are the major error types and feedback types encountered during annotation and verification?

G. Why use instructGPT instead of stronger LLMs like ChatGPT and GPT-4?

**Reasons To Accept:**

The motivation for the incorporation of ToM is crucial for narrative understanding and ToM serves as a suitable intermediate step for CoT reasoning.

The dataset is carefully collected and could be useful for future research in human-centric video and narrative comprehension.

The findings indicate that ToM-enhanced models yield superior performance based on automatic measures.

**Reasons To Reject:**

The size of the dataset is relatively small, restricting its use primarily for evaluation purposes. Also, its reliance on CSI content limits its applicability to a specific genre related to crime instead of daily life scenarios.

Some aspects of the annotation/experiments design require further clarification.

The experiments lack systematic human evaluation for model performance. It's not clear if the small difference in automatic evaluation measures can drive meaningful conclusions.

**Reproducibility:**

4: Could mostly reproduce the results, but there may be some variation because of sample variance or minor variations in their interpretation of the protocol or method.

**Reviewer Confidence:**

4: Quite sure. I tried to check the important points carefully. It's unlikely, though conceivable, that I missed something that should affect my ratings.

---

> ### Author Rebuttal · Authors · 2023-08-28
>
> We appreciate your constructive feedback!
>
> > The experiments lack systematic human evaluation for model performance. It's not clear if the small difference in automatic evaluation measures can drive meaningful conclusions.
>
> Thanks for highlighting the importance of systematic human evaluation.
> To validate whether the automatic metrics based on our annotated reference answers align with the actual quality of model predictions, we conduct additional human (on a subset) and GPT-4 evaluation (on the whole set) of the event causality inference task.
> For GPT-4 evaluation, we adapt the scoring framework from Zheng et al. (2023), with our annotated inferences as the reference answers, where scores range from $1$ to $10$.
> For human evaluation, we rank ($1$ to $4$) the model predictions on $10$\% ($238$ instances) of the dataset for simplicity.
> Specifically, we did the evaluation based on a set of criteria: correctness, relevance, completeness, and reasoning consistency (e.g., the extent to which the inference appropriately employs theory-of-mind).
> ~~&#10003;~~ represents the half-checkmark in Table 4.
>
> |   Models  |   VL   |   FS   |   CoT  |    ToM     |   Avg. Rank   |  GPT-4 Score  |
> |    :-:    |   :-:  |   :-:  |   :-:  |     :-:    |      :-:      |      :-:      |
> |InstructGPT|&#10003;|&#10003;|&#10007;|  &#10007;  |    $1.42$     |    $5.83$     |
> |InstructGPT|&#10003;|&#10003;|&#10003;|  &#10007;  |    $1.40$     |    $5.90$     |
> |InstructGPT|&#10003;|&#10003;|&#10003;|~~&#10003;~~|    $1.37$     |    $6.29$     |
> |InstructGPT|&#10003;|&#10003;|&#10003;|  &#10003;  |    $1.34$     |    $6.36$     |
>
> We observe the same trend of model performance on human ranking and GPT-4 scoring as the automatic metrics in Table 4.
> We will add the human and GPT-4 evaluation results on the whole dataset in our revised version.
>
> ---
> Judging LLM-as-a-judge with MT-Bench and Chatbot Arena. Zheng et al. 2023
>
> > **Q.A.** In Characteristics Identification, are different roles of the same character collected for each plot, or does a character's role remains consistent across most plots?
>
> Character roles are not static but can vary in different plots, depending on the nuances of their appearances, behaviors, and interactions in each specific context.
> For example, a character's role could be labeled based on their occupation, such as *forensic investigator*, or based on their specific actions within the plot, like *female passenger sitting next to the car driver*.
>
> > **Q.B.** In Keyframe-Grounded Emotion Interpretation, if multiple selections are allowed, how do annotators choose key frames for each emotion selection?
>
> The minimum number of frames to select is determined by both the clip duration and the number of selected emotions. When multiple emotions are identified, annotators are also instructed to select more frames.
> However, we didn't strictly enforce one-to-one matching between frames and emotions. This is because one frame can feature several emotions, and some emotions may be more accurately captured by considering the reactions of other characters in the scene.
> To ensure the completeness and informativeness of the selected keyframes, we have implemented a follow-up validation process that takes into account the roles and emotions annotated.
>
> > **Q.C.** In Event Causality Inference, are there any limitations on using utterances or narrations as events? How the selection of annotated "events" is determined?
>
> Thanks for the clarifying question on event determination.
>
> We select and determine the events through two stages. Initially, we randomly select utterances or narrations as events based on a set of constraints. These constraints include: 1) the event must mention character(s) in the plot, 2) the event should ideally occur in the middle of the plot to facilitate effective causality reasoning, and 3) the event is long enough to form a complete and understandable unit. Subsequently, in Round Two, the automatically selected events that are insufficiently meaningful are discarded. Annotators can reject such instances, pending confirmation from the authoring team.
>
> The main limitation of our automatic event determination is that it doesn't prioritize salience within the plot. This means that certain events, potentially more interesting or pertinent for causal reasoning, may go unselected.
>
> > **Q.D.** Sec 5 and Fig 4 are hard for me to understand. It can be helpful to clarify what intermediate steps are involved, what is the input and output of each step, and which model and prompt are used. There are too many prompts in Appendix D and it's hard for me to map them to each intermediate step. I also don't understand why some prompt requires a character name but some do not. How does the model know which person should be focused on?
>
> Thanks for the clarifying question.
>
> Denote the multimodal model and the language model as $M$ and $L$, respectively.
> The process for ToM-enhanced CoT reasoning for event causality inference is as follows (Steps 2 \& 3 can be merged for simplicity, with further post-processing to extract the final answers):
> 1. Visual Information Extraction using $M$: $M$ generates visual descriptions based on the input video clips. The prompt to $M$ determines the type and focus of the visual descriptions generated.
>     - input: clip frames
>     - output: visual descriptions
> 2. Role and Emotion Identification using $L$: We augment the text prompt for $L$ with the visual information obtained previously, eliciting roles and emotions associated with given characters.
>     - input: visual descriptions $+$ textual context (screenplay)
>     - output: roles and emotions of the specified characters
> 3. ToM-Enhanced CoT Reasoning using $L$: Using the predicted roles and emotions as intermediate reasoning products, we construct the CoT prompt for $L$, allowing it to perform ToM-enhanced reasoning about event causality.
>     - input: visual descriptions $+$ textual context (screenplay) $+$ roles and emotions
>     - output: event causality inference
>
> Regarding your question on prompts: In cases where the input includes screenplay contents, we specify character names to improve coreference resolution. For vision-only inputs, character identification can either be specific, pointing out characters featured in frames, or more general, referring to characters simply as "a person" when names are not needed.
>
> > **Q.E.** How to get the visual frames V in Sec 5? Are they provided by the annotated dataset or automatically selected from a video?
>
> The frames are manually selected following the role and emotion identification during annotation. There is also a follow-up step where annotators label the frames with the featured characters.
> We gather frames featuring different characters together to represent the visual content of each clip.
>
> > **Q.F.** How many instances are rejected and revised during inference validation? What are the major error types and feedback types encountered during annotation and verification?
>
> Out of the $5,746$ annotations collected from Round Two, we finally retained $4,280$ annotations, as indicated in Table 1.
> Among these, $107$ were revised and accepted following re-annotation.
> In validation, we check the annotation quality considering three criteria: plausibility, relevance, and completeness. Specifically, we mainly reject instances if they exhibit weak connections between the annotated inferences and the plot contents or the corresponding character roles and emotions.
>
> Feedback from annotators suggests two primary causes for annotation errors: 1) the imperfect event selection (where we reject the entire instance), and 2) insufficient incorporation of ToM in reasoning (where we reject outright or send it back for re-annotation).
>
> > **Q.G.** Why use instructGPT instead of stronger LLMs like ChatGPT and GPT-4?
>
> We use InstructGPT considering the reproducibility of model performance, as stronger closed-source LLMs like ChatGPT and GPT-4 will be updated periodically.
> For reference, we conduct an additional experiment using ChatGPT (`gpt-3.5-turbo-0613`) on the task of event causality inference.
>
> |   Models  |   VL   |   FS   |   CoT  |    ToM     |  BERT-F1  |  GPT-4 Score  |
> |    :-:    |   :-:  |   :-:  |   :-:  |     :-:    |     :-:   |     :-:       |
> |InstructGPT|&#10003;|&#10003;|&#10003;|~~&#10003;~~|  $63.18$  |    $6.29$     |
> |InstructGPT|&#10003;|&#10003;|&#10003;|  &#10003;  |  $63.65$  |    $6.36$     |
> |  ChatGPT  |&#10003;|&#10003;|&#10003;|  &#10003;  |  $64.09$  |    $6.98$     |
>
> > Some aspects of the annotation/experiments design require further clarification.
>
> Thanks for your above clarifying questions. Please see our responses above. We will revise our paper accordingly in the next version.
>
> > The size of the dataset is relatively small, restricting its use primarily for evaluation purposes; its reliance on CSI content limits its applicability to a specific genre related to crime instead of daily life scenarios.
>
> Thanks for raising your concerns on the size and scope of our dataset ECHo.
> We acknowledge that ECHo only represents a small subset of social interactions, but we view it as an initial and specific step to explore ToM understanding of social intelligence. As outlined in Table 2, crime scenes are particularly rich in ToM-related cases, including *false-belief* and *unexpected-content*.
> In light of your comment, would it enhance the clarity and utility if we further emphasize the specialized focus on crime scenes as a distinguishing aspect of our dataset?
> In this case, we will elaborate more on how ECHo assesses agents' ToM understanding in a specific genre, like humor detection (Patro et al. 2021).
>
> ---
> Multimodal Humor Dataset: Predicting Laughter tracks for Sitcoms. Patro et al. WACV 2021

---

### Official Review · Reviewer_Jxpn · 2023-08-05

**Soundness:** 3

**Excitement:**

4: Strong: This paper deepens the understanding of some phenomenon or lowers the barriers to an existing research direction.

**Paper Topic And Main Contributions:**

The authors propose ECHo, which is a visio-linguistic corpous for social reasoning. They further propose a framework to evaluate large language models in zero- and few-shot theory of mind enhanced reasoning. They further show how to leverage existing LLMs and multimodal models to conduct reasoning on ECHo

**Reasons To Accept:**

1. The paper is well-written and easy to follow.

2. The proposed dataset presents a way to evaluate artificial agents' understanding of Theory of Mind, which is crucial for implementing general artificial intelligence. I believe the contribution made here is solid.

3. The authors proposes a unified method (combining visual and linguistic reasoning) to conduct chain-of-thought reasoning on the proposed data.

**Reasons To Reject:**

1. The annotation of ECHo Corpus is a difficult task, which makes obtaining high quality data hard. I believe it is necessary here to justify the quality of the data. Although the authors present data construction details, they did not give quantitative evaluation for the annotation quality (e.g., inter-rater agreement in the classification tasks ). If the authors could not justify the quality of the proposed data, the later-mentioned experimental results could be less convincing.


**Reproducibility:**

4: Could mostly reproduce the results, but there may be some variation because of sample variance or minor variations in their interpretation of the protocol or method.

**Reviewer Confidence:**

3: Pretty sure, but there's a chance I missed something. Although I have a good feel for this area in general, I did not carefully check the paper's details, e.g., the math, experimental design, or novelty.

---

> ### Author Rebuttal · Authors · 2023-08-28
>
> We acknowledge your insightful comments!
>
> > The annotation of ECHo Corpus is a difficult task, which makes obtaining high quality data hard. I believe it is necessary here to justify the quality of the data. Although the authors present data construction details, they did not give quantitative evaluation for the annotation quality (e.g., inter-rater agreement in the classification tasks).
>
> Thanks for highlighting the importance of providing inter-rater agreement.
>
> We provide the inter-annotator agreement in Tables 3 and 4 in our submission. The automatic scoring is designed to treat annotator inputs as a reference to each other for each instance. For example, the macro F1 of the emotion interpretation (multi-label classification) task is $75.36$. We will make this clear in the next version of our paper.
>
> To validate whether the automatic metrics based on our annotated reference answers align with the actual quality of model predictions, we conduct additional human (on a subset) and GPT-4 evaluation (on the whole set) of the event causality inference task.
> For GPT-4 evaluation, we adapt the scoring framework from Zheng et al. (2023), with our annotated inferences as the reference answers, where scores range from $1$ to $10$.
> For human evaluation, we rank ($1$ to $4$) the model predictions on $10$\% ($238$ instances) of the dataset for simplicity.
> Specifically, we did the evaluation based on a set of criteria: correctness, relevance, completeness, and reasoning consistency (e.g., the extent to which the inference appropriately employs theory-of-mind).
> ~~&#10003;~~ represents the half-checkmark in Table 4.
>
> |   Models  |   VL   |   FS   |   CoT  |    ToM     |   Avg. Rank   |  GPT-4 Score  |
> |    :-:    |   :-:  |   :-:  |   :-:  |     :-:    |      :-:      |      :-:      |
> |InstructGPT|&#10003;|&#10003;|&#10007;|  &#10007;  |    $1.42$     |    $5.83$     |
> |InstructGPT|&#10003;|&#10003;|&#10003;|  &#10007;  |    $1.40$     |    $5.90$     |
> |InstructGPT|&#10003;|&#10003;|&#10003;|~~&#10003;~~|    $1.37$     |    $6.29$     |
> |InstructGPT|&#10003;|&#10003;|&#10003;|  &#10003;  |    $1.34$     |    $6.36$     |
>
> We observe the same trend of model performance on human ranking and GPT-4 scoring as the automatic metrics in Table 4.
> We will add the human and GPT-4 evaluation results in our revised version.
>
> ---
> Judging LLM-as-a-judge with MT-Bench and Chatbot Arena. Zheng et al. 2023

---

### Official Review · Reviewer_DSz5 · 2023-08-05

**Soundness:** 3

**Excitement:**

4: Strong: This paper deepens the understanding of some phenomenon or lowers the barriers to an existing research direction.

**Paper Topic And Main Contributions:**

The paper aims to probe human-centric social intelligence and assess the reasoning ability of large foundation models in multiple modalities. The contributions of the paper are (1) introducing a visio-linguistic dataset of event causality inference, (2) proposing a chain of thought framework to assess reasoning of AI systems, using large foundation models, and (3) examining models by performing human-centric tasks. Specifically, the dataset is a visio-linguistic dataset with theory of mind inferences for event causality reasoning in social scenarios, with grounding in a plot from crime drama CSI: Crime Scene Investigation. The annotation process involves (1) identifying a specified character as their role, (2) discerning mental states via emotion interpretation, and (3) inferring cause or effect of plot event for causality reasoning.

**Questions For The Authors:**

Is the annotation line of role identification, emotion identification, and event causality identification supposed to be a pipeline? In some cases, event causality is not related to emotions. For example, in Figure 2, for the question what has probably caused / enabled the event that Holly gets out of the car? The answer is Holly, as a junior, listened to Grissom's advice... This is not very related to emotions.

How is evaluation impacted when multiple causes of an event may exist?

How do you deal with non-deterministic outputs from the model (even a temperature of 0 cannot guarantee a consistent output) and how often is non-determinism an issue?

Who are the annotators and what are their background (are they all similar)? What is the interannotator agreement?

**Reasons To Accept:**

The problem statement is very clear and the paper has many clear figures to explain the annotation pipeline, dataset statistics, examples of model inputs and outputs, and a thorough evaluation. The appendix has clear figures to describe the process of collecting annotations and of training annotators. Probing social intelligence over multiple modalities is a very practical and interesting problem and a dataset resource to help with this problem is very helpful. The dataset is on crime scenes which facilitate approximation of social interaction.

**Reasons To Reject:**

The fact that multiple different causes of an event can exist complicates the evaluation of the CoT approach for event causality.

The relationship between event causality and the previous step of the task pipeline (emotion identification) is not completely clear.

In crime scenes, some emotions may be more common than others (e.g. humor is not common in crime scenes), so evaluating on crime scene data may not be the most complete way to accomplish the larger goal of probing social intelligence.

**Reproducibility:**

4: Could mostly reproduce the results, but there may be some variation because of sample variance or minor variations in their interpretation of the protocol or method.

**Reviewer Confidence:**

2: Willing to defend my evaluation, but it is fairly likely that I missed some details, didn't understand some central points, or can't be sure about the novelty of the work.

**Typos Grammar Style And Presentation Improvements:**

On page 4, the second column has a typo of "for to".

In round three: inference validation -- describe more details about how to reject annotations.

Explain the reason that the pipeline is split into the parts of role identification, emotion identification, and event causality, or if all of these parts are supposed to be parallel, not dependent on, each other.

---

> ### Author Rebuttal · Authors · 2023-08-28
>
> We recognize your thoughtful feedback!
>
> > The fact that multiple different causes of an event can exist complicates the evaluation of the CoT approach for event causality. How is evaluation impacted when multiple causes of an event may exist?
>
> Thanks for highlighting the open-endedness of event causality inference.
> We dealt with the one-to-many problem on two aspects of data collection. First, we provide multiple reference inferences (annotations) for each event, as shown in Table 1. Second, we narrow the scope of potential causes by employing **Theory-of-Mind** (ToM) as the intermediate reasoning step. This serves as a directional constraint to reduce the search space of potential CoT chains.
>
> Considering that our annotated reference answers may not capture the full spectrum of possible causes, we conduct human evaluations on $10$\% ($238$ instances) of the dataset to **rank** ($1$ to $4$) the plausibility of the model's predictions independently.
> Specifically, the ranking is based on a set of criteria: correctness, relevance, completeness, and reasoning consistency (e.g., the extent to which the inference appropriately employs theory-of-mind). ~~&#10003;~~ represents the half-checkmark in Table 4.
>
> |   Models  |   VL   |   FS   |   CoT  |    ToM     |   Avg. Rank   |
> |    :-:    |   :-:  |   :-:  |   :-:  |     :-:    |      :-:      |
> |InstructGPT|&#10003;|&#10003;|&#10007;|  &#10007;  |    $1.42$     |
> |InstructGPT|&#10003;|&#10003;|&#10003;|  &#10007;  |    $1.40$     |
> |InstructGPT|&#10003;|&#10003;|&#10003;|~~&#10003;~~|    $1.37$     |
> |InstructGPT|&#10003;|&#10003;|&#10003;|  &#10003;  |    $1.34$     |
>
> The human evaluation ranks the model predictions in the same order as that in Table 4. Furthermore, the performance gain brought by ToM-constrained CoT shows the importance of ToM in our human-centric reasoning task.
> We will add the human evaluation results on the whole dataset in our revised version.
>
> > The relationship between event causality and the previous step of the task pipeline (emotion identification) is not completely clear. Is the annotation line of role identification, emotion identification, and event causality identification supposed to be a pipeline? In some cases, event causality is not related to emotions. For example, in Figure 2, for the question what has probably caused / enabled the event that Holly gets out of the car? The answer is Holly, as a junior, listened to Grissom's advice... This is not very related to emotions.
>
> Thanks for the clarifying question.
> Yes, we conduct the annotations of role, emotion, and event causality identification sequentially. Annotators are asked to refer to the annotated role(s) and/or emotion(s) when inferring the event causality. This process encourages Theory-of-Mind reasoning for human-centric plot understanding.
> Take Figure 2 for example, while Holly’s action of *getting out of the car* is not directly caused by her emotion (e.g., *entrancement*, *doubt*), it benefits understanding her decision as a junior (her role) who lacks experience of independent investigation.
> In the next version, we will further clarify the task relationships in the annotation pipeline.
>
> > In crime scenes, some emotions may be more common than others (e.g. humor is not common in crime scenes), so evaluating on crime scene data may not be the most complete way to accomplish the larger goal of probing social intelligence.
>
> Thanks for raising the incredibly important issue of probing social intelligence in a **complete** way.
> We acknowledge that our dataset ECHo represents a subset of social interactions, but we view it as an initial and specific step to explore ToM understanding of social intelligence. As outlined in Table 2, crime scenes are particularly rich in ToM-related cases, including *false-belief* and *unexpected-content*.
> In light of your comment, would it enhance the clarity and utility if we further emphasize the specialized focus on crime scenes as a distinguishing aspect of our dataset?
> In this case, we will elaborate more on how ECHo assesses agents' ToM understanding in a specific genre, like humor detection (Patro et al. 2021).
>
> ---
> Multimodal Humor Dataset: Predicting Laughter tracks for Sitcoms. Patro et al. WACV 2021
>
> > How do you deal with non-deterministic outputs from the model (even a temperature of 0 cannot guarantee a consistent output) and how often is non-determinism an issue?
>
> Thanks for raising your concern about the non-determinism of model predictions.
> We conduct additional experiments to probe how model performance varies with different temperatures.
> For reference, we also provide the GPT-4 scoring results based on our annotated inferences.
> We adapt the scoring framework from Zheng et al. (2023), where scores range from $1$ to $10$.
>
> |   Models  |Temperature|   VL   |   FS   |   CoT  |    ToM     |  BLeU-2  |  Rouge-L  |  BERT-F1  |  GPT-4 Score  |
> |    :-:    |   :-:     |   :-:  |   :-:  |   :-:  |     :-:    |   :-:    |    :-:    |     :-:   |     :-:       |
> |InstructGPT|   $0.0$   |&#10003;|&#10003;|&#10003;|~~&#10003;~~| $10.63$  |  $22.20$  |  $63.18$  |    $6.29$     |
> |InstructGPT|   $0.0$   |&#10003;|&#10003;|&#10003;|~~&#10003;~~| $10.04$  |  $22.03$  |  $62.96$  |    $6.30$     |
> |InstructGPT|   $0.5$   |&#10003;|&#10003;|&#10003;|~~&#10003;~~|  $9.91$  |  $21.72$  |  $62.65$  |    $6.25$     |
> |InstructGPT|   $0.0$   |&#10003;|&#10003;|&#10003;|  &#10003;  | $11.28$  |  $22.97$  |  $63.65$  |    $6.36$     |
> |InstructGPT|   $0.0$   |&#10003;|&#10003;|&#10003;|  &#10003;  | $11.12$  |  $22.80$  |  $63.43$  |    $6.42$     |
> |InstructGPT|   $0.5$   |&#10003;|&#10003;|&#10003;|  &#10003;  |  $9.65$  |  $22.02$  |  $63.08$  |    $6.37$     |
>
> Our results indicate minimal variation in performance when using different temperatures, in terms of both automatic metrics and GPT-4 scores. For instance, the GPT-4 score fluctuated only within a narrow range of $6.25$ to $6.42$ across different settings. So non-determinism does not significantly impact the model's output quality.
>
> ---
> Judging LLM-as-a-judge with MT-Bench and Chatbot Arena. Zheng et al. 2023
>
> > Who are the annotators and what are their background (are they all similar)? What is the interannotator agreement?
>
> Thanks for your clarifying questions about our annotator background and the inter-annotator agreement.
>
> We recruited annotators considering two-fold criteria. First, to ensure the annotation quality, all selected annotators possess a minimum of undergraduate education and demonstrate familiarity or expertise with the TV series *CSI*. Second, to control for potential biases and ensure diversity, our recruitment aims for a balanced distribution across various factors, including but not limited to gender, academic disciplines, and nationalities.
>
> We provide the inter-annotator agreement in Tables 3 and 4 in our submission. The automatic scoring is designed to treat annotator inputs as a reference to each other for each instance. For example, the macro F1 of the emotion interpretation task is $75.36$.
> We will make this clear in the next version of our paper.
>
> > **Presentation Improvement**: In round three: inference validation -- describe more details about how to reject annotations.
>
> Thanks for your suggestion.
>
> The inference validation process consists of two parts: automatic and manual validation.
> The real-time automatic checking alerts annotators if their inputs fail to meet certain informativeness criteria, such as text length and word-level overlap with the existing context.
> Subsequently, our authoring team carry out the manual validation. We particularly focus on instances with lower inter-rater agreement in emotion identification or anomalies in the timestamp distribution of selected frames. We evaluate annotations based on three dimensions: plausibility, relevance, and completeness. For example, we reject instances where the event causality inference shows weak association with either the plot or the annotated roles and emotions.
> Based on our $5,746$ annotations from Round Two, we finally collect $4,280$ annotations (Table 1) after rejection and revisement.
>
> We will further clarify this in the Section of construction pipeline (Section 3.1) in our paper.
>
> > **Presentation Improvement**: Explain the reason that the pipeline is split into the parts of role identification, emotion identification, and event causality, or if all of these parts are supposed to be parallel, not dependent on, each other.
>
> Thanks for your suggestion.
>
> We conduct the role, emotion, and event causality identifications sequentially in our annotation pipeline. Specifically, the event causality inference is dependent on the prior annotations concerning roles and emotions. These tasks aim to assess social intelligence through foundational aspects of human-centric reasoning, i.e., ToM understanding. Based on these, the task of event causality inference further examines reasoning capacities within socially-grounded interactions.
>
> We will further clarify this in the Section of task formulation (Section 4) in our revised version.
>
> > **Typo**: On page 4, the second column has a typo of "for to".
>
> We appreciate the reviewer’s correction of the typo and the above suggestions on presentation. We will revise our paper accordingly in the next version.

---

### Meta-Review · Area_Chair_Wk8U · 2023-09-12

**Recommendation:** 4

**Metareview:**

The paper presents a multimodal dataset of theory of mind inferences for event causality reasoning in social scenarios, which are set in a crime setting. The paper shows that theory-of-mind enhanced models achieve better performance on the task that models that do not perform such reasoning.
The reviewers agree that the presented task is an important and timely task, and that the dataset provides a new unique resource for this task; they also agree that the paper is presented very clearly, that the dataset is described well and that the evaluation is thorough. The concerns of the reviewers pertain to the generalizability of findings  / distributions of social inferences in the particular crime domain that was chosen to construct the dataset, but I think that the authors can address this in the discussion in the final version of the paper. The reviewers also had some questions regarding annotator agreement and a human evaluation, which are addressed well in the rebuttal, in my opinion.

---

### Decision · Program_Chairs · 2023-10-07

**Decision:**

Accept-Findings

**Comment:**

The paper presents a multimodal dataset of theory of mind inferences for event causality reasoning in social scenarios, which are set in a crime setting. The paper shows that theory-of-mind enhanced models achieve better performance on the task that models that do not perform such reasoning.
The reviewers agree that the presented task is an important and timely task, and that the dataset provides a new unique resource for this task; they also agree that the paper is presented very clearly, that the dataset is described well and that the evaluation is thorough. The concerns of the reviewers pertain to the generalizability of findings  / distributions of social inferences in the particular crime domain that was chosen to construct the dataset, but I think that the authors can address this in the discussion in the final version of the paper. The reviewers also had some questions regarding annotator agreement and a human evaluation, which are addressed well in the rebuttal, in my opinion.